# A Comparison of Developmental Profiles of Preschool Children with Down Syndrome, Global Developmental Delay, and Developmental Language Disorder

**DOI:** 10.3390/healthcare13141684

**Published:** 2025-07-13

**Authors:** Mónica-Alba Ahulló-Fuster, M. Luz Sánchez-Sánchez, Alejandro Monterrubio-Gordón, Maria-Arantzazu Ruescas-Nicolau

**Affiliations:** 1Department of Radiology, Rehabilitation and Physiotherapy, Faculty of Nursing, Physiotherapy and Podiatry, Complutense University of Madrid, 28040 Madrid, Spain; mahullo@ucm.es; 2Physiotherapy in Motion, Multispeciality Research Group (PTinMOTION), Department of Physiotherapy, University of Valencia, 46010 Valencia, Spain; arancha.ruescas@uv.es; 3Department of Physiotherapy, Faculty of Physiotherapy, Universitat de València, 46010 Valencia, Spain

**Keywords:** BDI−2, developmental milestones, Down syndrome, developmental language disorder, global developmental delay, developmental disability, atypically developing toddlers

## Abstract

**Background/Objectives:** Developmental disabilities substantially affect the daily lives of children and their families. Although interest in examining the developmental profiles of children with various disabilities has grown, few studies have systematically compared them. This study aimed to characterize the developmental profiles of preschool-aged children with different disabilities. It was hypothesized that developmental profiles would differ depending on the type of developmental disability. **Methods**: A cross-sectional study was conducted. Scores on the Battelle^®^ Developmental Inventory, 2nd Edition (BDI−2) were retrieved for a non-probabilistic convenience sample of 46 children diagnosed with Down syndrome (DS) (*n* = 22), global developmental delay (GDD) (*n* = 17), and developmental language disorder (DLD) (*n* = 7) upon completion of an early intervention program. Developmental quotients (DQs) for the overall BDI−2 and for each domain were determined. **Results**: The children’s mean age was 42.39 ± 5.23 months (range: 30–57). Significant differences were observed among groups with regard to global DQ and all domain-specific DQs (*p* ≤ 0.01). The GDD group demonstrated the highest DQs across all domains and globally, in comparison to the other groups. Conversely, children with DS had substantially lower DQs across all domains and globally compared to those with GDD, and in the motor and communication domains compared to children with DLD. **Conclusions**: These findings underscore the importance of early intervention strategies to improve communication in children with DS and highlight the need for regular assessments to monitor progress and identify potential limitations, particularly during the preschool-to-school transition. Additionally, specialists should advise parents of children with DLD to adopt specific behaviors that support the development of their children’s social, adaptive, and language skills.

## 1. Introduction

The initial 5 years of life, often designated as early childhood, represent a critical period during which the developing brain is highly receptive to external stimuli and novel learning experiences [1,2]. This heightened sensitivity promotes rapid maturation and development, thereby shaping the child’s overall growth [1]. However, during this developmental stage, substantial delays in typical developmental processes can occur, potentially resulting in developmental disabilities [2]. Globally, it is estimated that approximately 5 million children aged 0–5 years live with moderate to severe disabilities [3]. The prevalence of these conditions, however, varies substantially across different geographical regions. Developmental disabilities are more commonly reported in low- and middle-income countries than in high-income ones [3,4].

Developmental disabilities have a substantial impact on the lives of affected children, resulting in functional limitations across multiple areas due to delays in achieving key developmental milestones [4]. These delays may be caused by various childhood conditions, most frequently including Down syndrome (DS), neuropsychomotor or global developmental delay (GDD), and developmental language disorder (DLD) [5]. DS arises from congenital abnormalities and is the most common aneuploid condition present at birth [5,6,7]. GDD is a relatively common developmental disorder for which no underlying cause is readily apparent [8,9], and it typically involves delays in at least two developmental domains [10,11]. DLD, by contrast, is characterized by impaired expressive and/or receptive language skills, despite normal hearing and appropriate non-verbal intelligence [5,6,7,12,13]. It occurs in the absence of neurological damage, structural oral anomalies, or other medical conditions [12,13,14,15], and is most frequently diagnosed in children under 5 years of age [16].

In addition, it should be noted that despite their distinct etiologies and presenting characteristics, the neurodevelopmental profiles of children with these conditions exhibit substantial variability [17]. The manifestation of delays in these developmental domains often impedes the child’s ability to engage in daily activities, exerting a detrimental effect on both the child and their family and community. This, in turn, increases the child’s dependence on caregivers, thereby compromising their overall quality of life [18,19]. Consequently, the early detection and timely rehabilitation of children with developmental disabilities is of paramount importance [4].

Despite the availability of early interventions, the global burden of developmental disabilities has shown limited improvement since the 1990s [19]. This underscores the need for a comprehensive analysis and comparison of the developmental profiles of children with such disabilities. The scientific literature includes studies examining the developmental profiles of children with various medical conditions, such as severe acute malnutrition [20], middle cerebral artery infarction, periventricular infarction [21], and autism spectrum disorder [22]. Nevertheless, analyzing developmental profiles across different medical conditions is important, as children with varying conditions may respond differently to rehabilitation interventions [21]. Such an approach may offer a broader perspective on common challenges and highlight inadequacies in current intervention strategies.

However, research examining and comparing the developmental profiles of children with different developmental disabilities remains limited. Many studies focus on only one or two groups of children with disabilities, or divide children with the same diagnosis into subgroups [14,15,16,17], and only a few compare multiple groups with diverse developmental conditions [23]. Moreover, existing studies tend to focus on a limited number of developmental areas [24,25,26], overlooking the importance of thoroughly evaluating multiple domains. Furthermore, most research in this field centers on newborns, school-aged children, or preadolescents [25,26,27,28], with relatively few studies involving preschool-aged children who have received therapeutic intervention in an early intervention center (EIC) [23].

To address this gap, the primary objective of this study was to analyze the developmental profiles of preschool-aged children with various disabilities upon completion of an EIC program. Specifically, a comparison was made between the developmental profiles of children diagnosed with DS, GDD, and DLD. These profiles were assessed using the Battelle^®^ Developmental Inventory, 2nd Edition (BDI−2). For the purpose of this study, developmental quotients (DQs) for the complete inventory and for each domain were determined. It was hypothesized that children with different developmental disorders would demonstrate distinct developmental profiles, depending on their specific diagnoses.

## 2. Materials and Methods

### 2.1. Design and Participants

The present study adopted a retrospective cross-sectional study design. Participants were obtained from Asindown, an association of people with developmental delays in Valencia, Spain. The clinical records of children who attended the EIC of Asindown were retrieved over a period of 3 months. Consequently, non-probabilistic convenience sampling was utilized. The statistical power of the study was subsequently assessed post hoc using G*Power v3.1 software. The power was determined to be 99.9% at a 0.05 Type I error (two-tailed), with an effect size of 1.07 (η_p_^2^ = 0.536 derived from the comparison of global DQ among groups using a one-way analysis of variance [ANOVA]; please, refer to the results section), and a sample size of 46 participants.

Eligibility for the records under scrutiny was contingent upon the following criteria: (1) children with isolated diagnoses of DS, GDD, or DLD at the time of enrollment in the EIC; (2) attendance of an early intervention program during preschool years [29]; and (3) a developmental assessment using the second edition of the BDI−2 prior to completion of the EIC. Records were excluded if they (1) included multiple concurrent diagnoses (e.g., DS and GDD); (2) presented any comorbidities; (3) were incomplete; or (4) did not include all BDI−2 domains.

Written informed consent was obtained by the center from all participants involved in the study for data extraction, and a confidentiality agreement was signed with Asindown. The clinical records never left the EIC. To ensure anonymity and protect participants’ identities, a numerical code was assigned to each participant. The study was conducted in full compliance with the Declaration of Helsinki [30] and was approved by the authors’ institutional ethics committee (approval number 1794786). Additionally, the report adheres to the Strengthening the Reporting of Observational Studies in Epidemiology (STROBE) guidelines [31].

### 2.2. Procedures

A researcher regularly visited the EIC to extract data over a period of 3 months. Information on children who attended the early intervention program was collected in an Excel spreadsheet. Clinical and demographic data (sex, age in months, ethnicity, and clinical diagnosis), as well as developmental functioning (BDI−2), were retrieved from medical records.

The BDI−2 [32] is a widely used, standardized instrument designed to assess key developmental skills through observation, supplemented by parental report, in children from birth to 7 years and 11 months. It is a discriminative norm-referenced measure of developmental activities. It identifies strengths and weaknesses in children’s developmental areas to help create personalized intervention programs. In the collaborating EIC, prior to admission to primary school, the BDI−2 was administered by members of the early intervention team, who were experienced in assessing children with disabilities and trained in BDI−2 administration. Specifically, a neuropsychologist assessed the personal/social and cognitive domains, a physiotherapist the motor domain, a speech and language therapist the communication domain, and an occupational therapist the adaptive domain. Depending on each child’s individual characteristics, each professional required between 15 and 60 min to assess the corresponding domain. For this study, data from this assessment were of interest.

The BDI−2 consists of 341 items grouped into five domains: (1) personal/social; (2) motor; (3) communication; (4) cognitive; and (5) adaptive [33]. Each item is scored on a 3-point scale (0: no ability in the skill, 1: emerging ability, 2: acquired skill) [33,34]. Item scores are summed to produce a domain score, and the sum of domain scores produces a global score. To allow for comparison among individuals of different ages, raw scores are converted to age equivalents using the normative scales provided by the inventory. Subsequently, domain-specific and global DQs were calculated using the following formula: age equivalent/chronological age × 100 [35]. The global DQ has a mean value of 100 and a standard deviation of 15 [32]. For the purpose of statistical analysis, DQs were used. The BDI−2 has demonstrated acceptable to excellent psychometric properties in terms of reliability and consistency, and its validity has been established for use across a range of developmental disorders [36]. It has also been used in studies similar to ours [23,37,38].

### 2.3. Statistical Analysis

Statistical analysis was performed using IBM SPSS Statistics software (v. 24, IBM Corp; Armonk, NY, USA). Data are presented as frequency (percentage) or mean ± standard deviation, as appropriate. The normality of continuous variables was assessed with the Shapiro–Wilk test. Sample characteristics were compared between groups using a one-way ANOVA and the chi-squared (χ^2^) test, as appropriate.

To evaluate differences between groups in the DQs of each domain and the global DQ of the BDI−2, several ANOVAs were conducted. To assess ANOVAs effect size, partial eta-squared (η_p_^2^) was calculated. Post hoc comparisons with Bonferroni correction were performed to identify differences between pairs of groups. Effect sizes according to Hedges’ g were also calculated to ensure comparability in the pairwise analyses.

Finally, to determine whether the global DQ of each group differed from the standard reference value (mean = 100), three one-sample t-tests were conducted (one per group). Cohen’s method was used to calculate the corresponding effect sizes. A *p*-value of <0.05 was considered the threshold for statistical significance.

## 3. Results

A total of 170 children were enrolled at the EIC at the time of this study. Of these children, 71 were not expected to be discharged to a primary school, 26 did not have a definitive diagnosis, 9 presented comorbidities, 8 were transferred to another center, and 3 were hospitalized. Therefore, 51 children were assessed using the BDI−2, but only 46 children (22 with DS, 17 with GDD, and 7 with DLD) met the inclusion criteria. Data from these 46 children were included in the study. Figure 1 illustrates the flow of participants.

The sample consisted of 32 male and 14 female children, with a mean age of 42.4 ± 6.1 months. All participants identified as Caucasian. Upon division of the sample into groups (Table 1), it was observed that the three groups (DS, GDD, and DLD) did not differ significantly in terms of age (F_(2, 43)_ = 1.765, *p* = 0.183). However, a statistically significant difference was identified among the groups with respect to sex (χ^2^_(2)_ = 6.170, *p* = 0.046). Notably, the DLD group consisted exclusively of male participants (Table 1). Nevertheless, the association between sex and clinical diagnosis was weak (*C* = 0.29, *p* = 0.122).

Data are expressed as mean ± standard deviation or frequency (percentage). Between-group differences were calculated by using a one-way ANOVA for continuous data and the chi-squared test for categorical data. ES: effect size. Partial eta-squared (η^2^_p_) was used for continuous variables and the contingency coefficient (C) for categorical variables. DS: Down syndrome; GDD: global developmental delay; DLD: Developmental Language Disorder.

The findings of the between-group comparisons of domain-specific DQs and the global DQ of the BDI−2 are shown in Table 2 and Figure 2. The ANOVAs revealed statistically significant differences for all studied variables, indicating the presence of differences among the three groups.

In the post hoc comparisons (Figure 2a), the mean global DQ of the GDD group was significantly higher than that of the DS group (t_(37)_ = 7.01, *p* < 0.001, g = 2.326) and the DLD group (t_(22)_ = 3.20, *p* = 0.008, g = 1.121). However, the DS and DLD groups did not differ significantly (t_(27)_ = 1.89, *p* = 0.193, g = −0.964).

The pairwise comparisons of domain-specific DQs showed that children with GDD had significantly higher scores than those with DS and DLD in the personal−social domain (t_(37)_ = 5.42, *p* < 0.001, g = 1.790; t_(22)_ = 3.93, *p* < 0.001, g = 1.424, respectively) (Figure 2b) and the adaptive domain (t_(37)_ = 3.93, *p* < 0.001, g = 1.287; t_(22)_ = 2.67, *p* = 0.03, g = 1.078, respectively) (Figure 2c). In the motor domain (Figure 2d), the DS group exhibited significantly lower DQs compared to both the GDD group (t_(37)_ = 6.78, *p* < 0.001, g = −2.130) and the DLD group (t_(27)_ = 3.07, *p* = 0.010, g = −1.442). In the communication domain (Figure 2e), children with DS demonstrated lower DQs compared to those with GDD (t_(37)_ = 8.32, *p* < 0.001, g = −2.915) and DLD (t_(27)_ = 3.58, *p* = 0.02, g = −2.309). In the cognitive domain (Figure 2f), the GDD group showed significantly higher mean DQs than the DS group (t_(37)_ = 6.65, *p* < 0.001, g = 2.383) and the DLD group (t_(22)_ = 2.57, *p* = 0.04, g = 0.956).

Comparison of the global mean DQ of the three groups with the BDI−2 normative value (mean = 100) revealed a significant difference in all three groups, as indicated by the one-sample t-test: DS group: t_(21)_ = 24.53, *p* < 0.001, r = 0.982; GDD group: t_(16)_ = 4.03, *p* = 0.001, r = 0.710; and DLD group: t_(6)_ = 5.59, *p* = 0.001, r = 0.915. All groups exhibited lower global DQs (Table 2) in comparison to the BDI−2 reference mean.

## 4. Discussion

The present study has shown that, as hypothesized, children with developmental disorders exhibit different developmental profiles depending on their specific diagnostic condition. This is consistent with findings from other studies comparing groups of children with different health diagnoses [23,37,39,40].

Our results showed that, compared to individuals with DS and DLD, children with GDD obtained the highest global and domain-specific DQs, scoring, on average, above 70 in some domains (personal/social and adaptive), above 80 in others (motor and communication), and 100 in the cognitive domain. Moreover, the GDD group achieved an average global DQ score of 84.25 ± 16.09, which was significantly higher than the average scores obtained by the DS (54.74 ± 8.65) and DLD (65.49 ± 16.33) groups. However, previous research has reported a lower developmental profile in children with GDD compared to, for example, those born prematurely [23] or children with DLD [37]. As Horowitz and Matson [39] and Sharma et al. [11] noted, differences in children’s developmental levels may be influenced by multiple factors, such as the etiology of GDD, the nature and extent of early intervention, or the age at the time of assessment. These factors could explain the discrepancies between previous findings and ours. Moreover, toddlers with GDD often experience delays across multiple domains [9], making the clinical presentation of GDD heterogeneous. Notably, these delays tend to be more severe and pronounced in specific developmental areas when children with GDD present additional comorbid conditions, such as autism spectrum disorder [41,42] or intellectual disability [43,44]. For this reason, some researchers highlight the need for genetic testing to determine the underlying nature of GDD [45,46]. It is widely accepted that GDD is characterized by diverse phenotypes and a complex etiology [46]. Given that additional pathologies were an exclusion criterion in this study, it is highly probable that the members of our GDD group were relatively more homogeneous in this regard.

Regarding differences between groups in specific developmental domains, participants diagnosed with DS in the present study exhibited the most deficient motor domain scores when compared with those diagnosed with DLD and GDD. This is consistent with findings from other studies [23,39,47,48]. Children with DS frequently experience motor delays, primarily due to hypotonia and ligament laxity, which contribute to joint instability and hinder motor development [49,50]. Consequently, they struggle to achieve key gross motor milestones such as sitting, standing, and walking [25,49], and often exhibit difficulties with fine motor skills, including manual dexterity and bimanual coordination [47,51]. Notably, motor difficulties may persist beyond childhood, even when early intervention has been provided [52,53].

Our results also showed that the DS group had greater impairments in the communication domain compared to the other two groups. While some studies suggest that children with DS exhibit communication skills similar to those of peers with other developmental disabilities [23,39,48], it is important to consider that previous research often assessed them at a younger age. However, language impairments in children with DS tend to become more evident during the transition from preschool to school age [23,54], which aligns with the developmental period in which our sample was evaluated. Therefore, our findings highlight the need to develop early intervention strategies for communication in children with DS, as well as the importance of regular assessments to monitor progress and detect potential limitations [55], especially during this critical period.

In children with DS, receptive language is typically more developed than expressive language, with expressive difficulties primarily affecting phonology and syntax [54,56]. Additionally, hearing impairments and the distinctive orofacial anatomy associated with DS further hinder speech perception and production [57]. Structural differences, such as an enlarged tongue and an underdeveloped maxilla, contribute to a higher incidence of articulatory errors in individuals with DS [58,59].

Another notable finding of this study was the difference in cognitive abilities among groups. The GDD group obtained substantially higher scores in the cognitive domain of the BDI−2 compared to the DS and DLD groups. While a previous study similar to ours [23] found no cognitive differences between individuals with DS and those with other developmental disabilities, it is well established that DS is associated with cognitive limitations. However, the scientific literature acknowledges that these limitations exhibit a degree of inter-individual variability, suggesting the presence of distinct cognitive profiles within this population [60,61]. Specific areas, such as memory [62] and executive functioning [63], tend to be particularly affected [64]. Furthermore, research on general intelligence suggests that children with DS develop at a slower pace than their typically developing peers [65]. In this regard, our results underscore the importance of addressing the cognitive aspects of DS through therapeutic interventions. Exercise interventions, in particular, have been shown to produce substantial enhancements in cognitive functioning in individuals with DS [66].

Additionally, the lower cognitive scores observed in the DLD group of our study are consistent with previous research showing that children with DLD experience alterations in cognitive skills [67,68,69]. Studies have identified difficulties across multiple cognitive do-mains, including visuospatial working memory [68,70], phonological short-term memory [71], executive functioning [67], and cognitive flexibility [72]. Consequently, the literature has extensively explored whether the core deficit in DLD is primarily cognitive or linguistic [73,74,75], highlighting the complexity of this condition.

In our study, significant differences were also observed in the personal/social and adaptive domains, with the DLD and DS groups obtaining lower scores compared to the GDD group. A recent systematic review found that children with DLD experience social difficulties, particularly in peer interactions such as turn-taking and game behavior [76], as well as challenges in social–emotional functioning [77]. Based on this, professionals may advise parents of children with DLD to adopt specific parenting practices—such as fostering and modeling positive relationships and supporting opportunities to practice social and adaptive skills—to help their children improve their abilities. This, in turn, could contribute to language development [78,79].

Likewise, toddlers with DS have been shown to exhibit limitations in social skills when compared to typically developing children [57,80]. In addition to social interactions, a child’s relationship with their environment is shaped by adaptive behavior, which supports increasing independence, purposeful engagement, and responsibility [81]. Consistent with previous research [82,83], our findings indicate that children with DS and DLD demonstrate reduced capabilities in adaptive behavior. However, some studies suggest that adaptive behavior in these groups may be either superior [37] or comparable to that of other developmental disabilities [23], underscoring the need for further research in this area.

Lastly, in this study, children—regardless of their specific pathology—exhibited lower global DQs compared to the normative reference values established in the BDI−2. This finding aligns with outcomes from previous studies [23,37,39,84]. Typically, the DQs of children with developmental disabilities are evaluated using composite scores derived from multiple developmental scales to determine the presence and extent of developmental delays. The persistence of these disabilities varies considerably depending on the condition. While some children may overcome their diagnosis through maturation and early intervention programs, others face limitations that are intrinsic to their pathology. This is particularly evident in genetic conditions [85,86,87], such as DS, where full recovery of certain functions is not feasible [88]. In contrast, children with DLD and GDD often show a more favorable prognosis [8,89]. Nevertheless, accurate and timely intervention and assessment are essential in all cases to achieve the best possible developmental outcomes for children with developmental disabilities.

### Limitations

There are several limitations to this study that should be acknowledged. The participants were drawn from a single center, which may have introduced selection bias. However, the requirements for access to an EIC remain consistent across all Spanish centers. Furthermore, the small sample size in the DLD group (*n* = 7) substantially limits the generalizability of the results and increases the risk of a Type II error. While this is a notable limitation, the current scientific literature on the developmental profiles of preschool-aged children with DLD remains sparse. In this context, the study contributes to an important area, as ensuring that children are ready to learn when they start school has become a key priority. However, the general education curriculum often does not accommodate preschool children with developmental delays [90]. Therefore, it is essential to identify developmental profiles at this stage, especially in frequently occurring medical conditions.

Additionally, several factors may have influenced our results. This was a retrospective analysis based solely on BDI−2 scores obtained either at the conclusion of early therapeutic intervention or just before EIC discharge. Consequently, future research should evaluate and compare children’s progress both upon entering and just before leaving the EIC to gain a more comprehensive understanding of the developmental trajectories of GDD, DLD, and DS. Moreover, our study did not explore the underlying causes of GDD in the participants. Future studies are encouraged to investigate developmental profiles in this population while considering the etiology of the disorder and other potential confounding factors that may influence children’s outcomes (e.g., sex, age, duration, and intensity of the intervention, socioeconomic status, level of family involvement, family environment, and concurrent therapies).

Due to the design of the study, the assessors were not blinded to the children’s diagnoses. However, the researcher who performed the statistical analysis was blinded. Lastly, as in previous studies [91], assessments were conducted by a multidisciplinary team experienced in evaluating children with disabilities and trained in administering the BDI−2. It should be noted that, due to the operational structure of the center, each domain of the BDI−2 was evaluated by a professional from the area most closely related to that domain (e.g., a physiotherapist evaluated the motor domain). As such, the same professional evaluated all children within the same domain.

## 5. Conclusions

The developmental profiles of children with developmental delays vary depending on their medical diagnosis. Children with GDD exhibit a more favorable developmental profile—both globally and across BDI−2 domains—compared to those with DLD or DS. Additionally, children diagnosed with DS show lower scores in the motor and communicative domains than those with GDD or DLD. However, all three groups present global DQ scores lower than the BDI−2 reference value, confirming the presence of developmental delay.

From a clinical perspective, the findings underscore the need to implement early intervention strategies targeting communication in children with DS, along with the importance of regular assessments to monitor progress and identify potential limitations, particularly during the transition from preschool to school. Specialists are also encouraged to counsel parents of children with DLD to adopt specific behaviors aimed at enhancing their children’s social and adaptive skills, with the potential to improve communication abilities.

These findings highlight the importance of assessing developmental profiles in children with different types of developmental disabilities, as this can guide the creation of personalized intervention strategies and support parental guidance, ultimately improving outcomes for these children. Future multicenter and longitudinal studies could corroborate and expand upon these findings by tracking progress from admission to discharge in a larger sample of EIC users.

## Figures and Tables

**Figure 1 healthcare-13-01684-f001:**
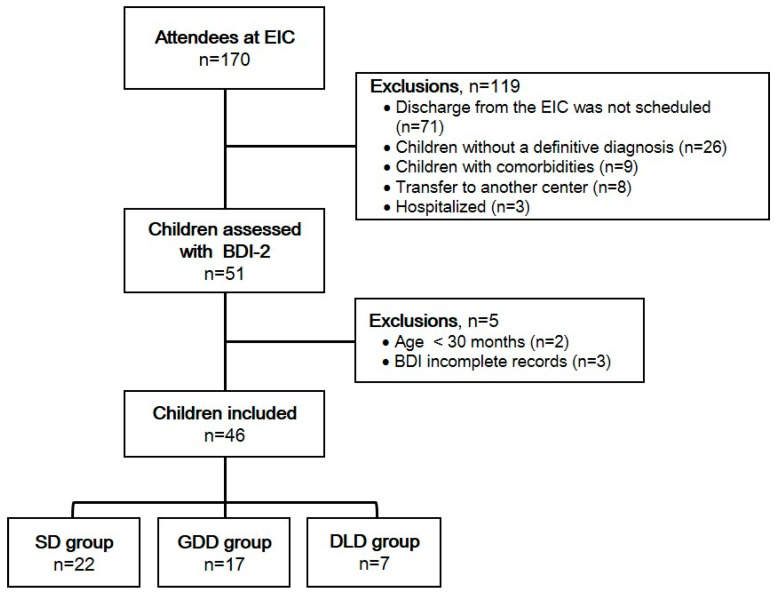
Flow diagram of participants. EIC: Early Intervention Center; BDI: Battelle Developmental Inventory, 2nd Edition; DS: Down syndrome; GDD: global developmental delay; DLD: Developmental Language Disorder.

**Figure 2 healthcare-13-01684-f002:**
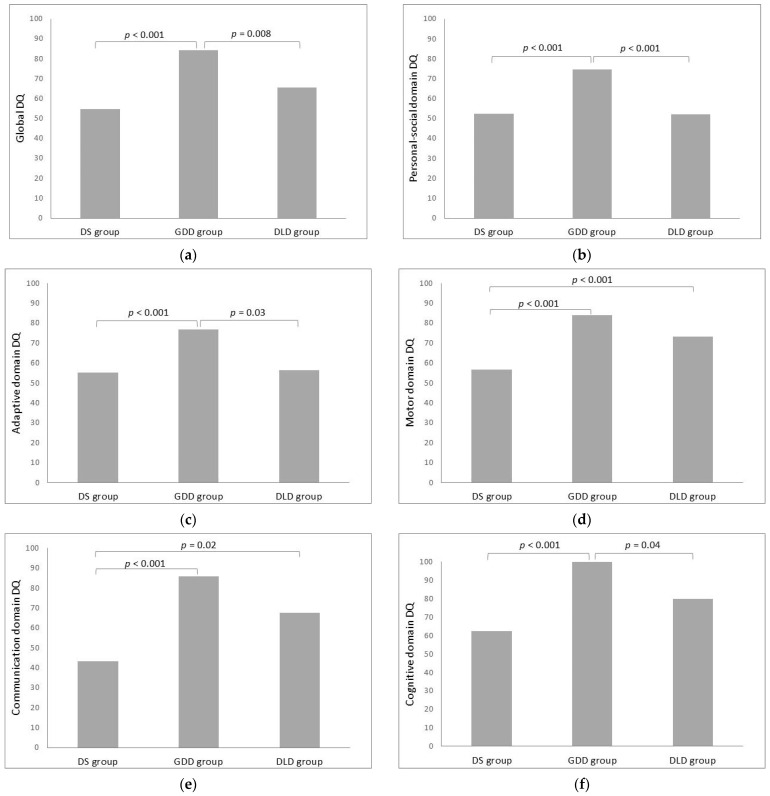
Results of the post hoc pairwise comparisons of the BDI−2 global DQ and specific-domain DQs: (**a**) global DQ; (**b**) personal–social domain DQ; (**c**) adaptative domain DQ; (**d**) motor domain DQ; (**e**) communication domain DQ; (**f**) cognitive domain DQ.

**Table 1 healthcare-13-01684-t001:** Sample characteristics by group.

	DS Group *N* = 22	GDD Group *n* = 17	DLD Group *n* = 7	*p* Value	ES
Age (months)	44.1 ± 5.5	40.6 ± 6.7	41.4 ± 5.1	0.183	0.076
Sex: Men Women	13 (59.1) 9 (40.9)	12 (70.6) 5 (29.4)	7 (100) 0 (0)	0.046	0.290
Ethnicity: Caucasian	22 (100)	17 (100)	7 (100)	1	-

**Table 2 healthcare-13-01684-t002:** Comparisons of the BDI−2 global and domain-specific developmental quotients (DQ) among groups.

	DS Group *n* = 22	GDD Group *n* = 17	DLD Group *n* = 7	Between-Group Differences		
				F_(2, 43)_	*p* Value	ES
Global DQ	54.74 ± 8.65	84.25 ± 16.09	65.49 ± 16.33	24.616	<0.001	0.534
Domain-specific DQs:						
Personal-social DQ	52.32 ± 9.44	74.66 ± 15.12	52.12 ± 15.69	16.472	<0.001	0.434
Adaptive DQ	55.51 ± 15.37	76.92 ± 17.42	56.67 ± 19.92	8.441	0.001	0.282
Motor DQ	56.82 ± 11.17	84.10 ± 14.15	73.40 ± 11.79	23.432	<0.001	0.521
Communication DQ	43.21 ± 7.22	85.87 ± 20.17	67.57 ± 17.05	40.179	<0.001	0.651
Cognitive DQ	62.48 ± 13.90	100.36 ± 17.52	79.98 ± 27.12	22.116	<0.001	0.507

DQ: Developmental quotient; DS: Down Syndrome; GDD: global developmental delay; DLD: Developmental Language Disorder; ES: partial η^2^ effect size statistics (η_p_^2^ = 0.01, small effect; η_p_^2^ = 0.06, medium effect; η_p_^2^ = ≥ 0.14, large effect).

## Data Availability

The data that support the findings of this study are available from the corresponding author, [MLSS], upon reasonable request.

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
