# Peer review of "A Comparison of Developmental Profiles of Preschool Children with Down Syndrome, Global Developmental Delay, and Developmental Language Disorder"

_healthcare, 2025, doi:10.3390/healthcare13141684_

Round 1

Reviewer 1 Report

Comments and Suggestions for Authors

This manuscript explores a highly relevant issue in the field of developmental pediatrics. It offers a comparative analysis of developmental trajectories among preschool children diagnosed with Down Syndrome (DS), Global Developmental Delay (GDD), and Specific Language Impairment (SLI), using the Battelle Developmental Inventory – Second Edition (BDI-2) as a standardized assessment tool. The study design, statistical analysis, and interpretation of findings are generally well executed. However, several methodological clarifications and editorial improvements are required before the manuscript can be considered for publication. 

The title accurately conveys the main objective of the study. Nonetheless, it should reflect updated clinical terminology: the term “Specific Language Impairment” (SLI) is outdated and should be replaced with “Developmental Language Disorder” (DLD), in accordance with recent diagnostic standards (e.g., CATALISE Consortium).

The abstract is well structured, but it is overly detailed in terms of statistical results. The inclusion of precise numerical data could be reduced in favor of highlighting the core findings and their implications. Additionally, the expression “retrospectively registered” is misapplied in this context and should be revised to “retrospectively collected” or simply “retrieved.”

The introduction provides an appropriate overview of the topic, supported by recent and relevant literature. However, the rationale for selecting these three specific diagnostic categories (DS, GDD, DLD) for comparative purposes should be made more explicit. It is also important to clarify whether the DLD group consisted exclusively of children with isolated diagnoses, without comorbidities, which would affect the validity of between-group comparisons. 

The retrospective cross-sectional design is methodologically suitable for the study objective. Nevertheless, the authors should use the term “non-probabilistic convenience sampling” rather than “convenient sample,” and should explicitly acknowledge the potential selection bias introduced by relying on data from a single Early Intervention Centre (EIC).

Regarding the administration of the BDI-2, additional details are needed: Were assessors blinded to the children's diagnoses? What was the training level of those administering each domain? The use of different professionals (e.g., physiotherapists for motor domains, speech therapists for others) should be justified, particularly from a psychometric reliability standpoint.

The statistical analysis is appropriately designed and executed. ANOVA, post-hoc comparisons, and effect sizes are correctly reported. For greater clarity and accessibility, the inclusion of visual data representations—such as bar graphs or radar charts comparing DQs across domains and groups—is strongly recommended.

The results are well organized and statistically robust. However, the small sample size in the DLD group (n = 7) significantly limits generalizability and increases the risk of Type II error. This should be explicitly acknowledged and discussed in greater depth. Additionally, Table 1 could benefit from the inclusion of effect size measures for categorical variables (e.g., Cramér’s V).

The discussion appropriately interprets the findings in the context of existing literature. Nevertheless, the clinical significance of statistically significant differences should be addressed more directly. The observation that children with GDD showed the highest DQs deserves deeper exploration, particularly considering the heterogeneity in outcomes reported by prior studies.

The use of language should be more precise—terms such as “more advanced developmental profile” should be replaced by specific references to the domains and comparative metrics involved. The use of the term DLD should be consistent throughout the discussion and aligned with international consensus.

The authors adequately identify key limitations, including the retrospective design and single-site data source. However, additional considerations should be incorporated:

  • The absence of longitudinal data tracking change over time.

  • The lack of control over potentially confounding variables (e.g., duration/intensity of intervention, socioeconomic status, level of family involvement).

These factors could have influenced the developmental outcomes reported and should be acknowledged accordingly.The conclusions are consistent with the study findings. However, a more nuanced interpretation is warranted, particularly in relation to clinical application and future research directions. The authors should consider recommending multicenter, longitudinal studies tracking progress from admission to discharge in early intervention settings to further validate and expand upon these findings.

Comments on the Quality of English Language

Minor typographical errors (e.g., “toe” instead of “to each” in line 107) should be corrected.

Author Response

We sincerely appreciate all the comments and suggestions for improvement of the manuscript provided by Reviewers 1, 2 and 3. Below, please find our point-by-point responses to each of the questions and suggestions. The changes in the manuscript are highlighted in yellow.

Reviewer 1

This manuscript explores a highly relevant issue in the field of developmental pediatrics. It offers a comparative analysis of developmental trajectories among preschool children diagnosed with Down Syndrome (DS), Global Developmental Delay (GDD), and Specific Language Impairment (SLI), using the Battelle Developmental Inventory – Second Edition (BDI-2) as a standardized assessment tool. The study design, statistical analysis, and interpretation of findings are generally well executed. However, several methodological clarifications and editorial improvements are required before the manuscript can be considered for publication.

  1. The title accurately conveys the main objective of the study. Nonetheless, it should reflect updated clinical terminology: the term “Specific Language Impairment” (SLI) is outdated and should be replaced with “Developmental Language Disorder” (DLD), in accordance with recent diagnostic standards (e.g., CATALISE Consortium).

Answer: The authors thank the reviewer for his/her valuable suggestion and agree with it. In line with this recommendation, the term 'Specific Language Impairment' has been changed to 'Developmental Language Disorder', in the title and throughout the document.

All of these changes are highlighted in yellow. Please refer to lines 4, 23, 37 and 56, among others.

  1. The abstract is well structured, but it is overly detailed in terms of statistical results. The inclusion of precise numerical data could be reduced in favor of highlighting the core findings and their implications. Additionally, the expression “retrospectively registered” is misapplied in this context and should be revised to “retrospectively collected” or simply “retrieved.”

Answer: We would like to thank the reviewer for his/her valuable comment. Following his/her advice, we have revised the abstract to highlight the core findings and their implications. Additionally, we have replaced the term 'retrospectively registered' to 'retrieved”.

Please refer to lines 31-36: “These findings underscore the importance of early intervention strategies to improve communication in children with DS and highlight the need for regular assessments to monitor progress and identify potential limitations, particularly during the pre-school-to-school transition. Additionally, specialists should advise parents of children with DLD to adopt specific behaviors that support the development of their children’s social, adaptive, and language skills.”

Please refer to lines 20-24: “Scores on the Battelle® Developmental Inventory, 2nd Edition (BDI-2) were retrieved for a non-probabilistic convenience sample of 46 children diagnosed with Down syndrome (DS) (n = 22), global developmental delay (GDD) (n = 17), and developmental language disorder (DLD) (n = 7) upon completion of an early intervention program.”  

  1. The introduction provides an appropriate overview of the topic, supported by recent and relevant literature. However, the rationale for selecting these three specific diagnostic categories (DS, GDD, DLD) for comparative purposes should be made more explicit. It is also important to clarify whether the DLD group consisted exclusively of children with isolated diagnoses, without comorbidities, which would affect the validity of between-group comparisons.

Answer: We would like to thank the reviewer for his/her valuable comments. Following his/her advice, we have rewritten part of the introduction to emphasize the rational of studying these three specific diagnostic categories. We would also like to point out that the material and methods section refers to the fact that one of the exclusion criteria was the presence of other developmental disabilities. We have rewritten this to make it clearer.

Please refer to lines 54-67: “These delays may be caused by various childhood conditions, most frequently includ-ing Down syndrome (DS), neuropsychomotor or global developmental delay (GDD), and developmental language disorder (DLD) [5]. DS arises from congenital abnormali-ties and is the most common aneuploid condition present at birth [5–7]. GDD is a rela-tively common developmental disorder for which no underlying cause is readily ap-parent [8,9], and it typically involves delays in at least two developmental domains [10,11]. DLD, by contrast, is characterized by impaired expressive and/or receptive language skills, despite normal hearing and appropriate non-verbal intelligence [5–7,12,13]. It occurs in the absence of neurological damage, structural oral anomalies, or other medical conditions [12–15], and is most frequently diagnosed in children under 5 years of age [16].

In addition, it should be noted that despite their distinct etiologies and presenting characteristics, the neurodevelopmental profiles of children with these conditions ex-hibit substantial variability [17].”

Please also refer to lines 113-118: “Eligibility for the records under scrutiny was contingent upon the following criteria: (1) children with isolated diagnoses of DS, GDD, or DLD at the time of enrollment in the EIC; (2) attendance of an early intervention program during preschool years [29]; and (3) a developmental assessment using the second edition of the BDI-2 prior to completion of the EIC. Records were excluded if they (1) included multiple concurrent diagnoses (e.g., DS and GDD); (2) presented any comorbidities; (3) were incomplete; or (4) did not include all BDI-2 domains.”

  1. The retrospective cross-sectional design is methodologically suitable for the study objective. Nevertheless, the authors should use the term “non-probabilistic convenience sampling” rather than “convenient sample,” and should explicitly acknowledge the potential selection bias introduced by relying on data from a single Early Intervention Centre (EIC).

Answer: We thank the reviewer for his/her kind suggestion. Following his/her advice, we have added the term 'non-probabilistic convenience sampling' for clarity. Additionally, the possible selection bias implied by relying on data from a single EIC has been acknowledged in the limitations section.

Please refer to lines 107-108: “Consequently, non-probabilistic convenience sampling was utilized”.

Please refer to lines 342-343: “The participants were drawn from a single center, which may have introduced selection bias.”

  1. Regarding the administration of the BDI-2, additional details are needed: Were assessors blinded to the children's diagnoses? What was the training level of those administering each domain? The use of different professionals (e.g., physiotherapists for motor domains, speech therapists for others) should be justified, particularly from a psychometric reliability standpoint.

Answer: We would like to thank the reviewer for his/her valuable comment. To clarify this point, we consulted with the early intervention center and expanded the information in the manuscript (methods and limitation sections).

Please refer to lines 138-145: “In the collaborating EIC, prior to admission to primary school, the BDI-2 was administered by members of the early intervention team, who were experienced in assessing children with disabilities and trained in BDI-2 administration. Specifically, a neuropsychologist assessed the personal/social and cognitive domains, a physiotherapist the motor domain, a speech and language therapist the communication domain, and an occupational therapist the adaptive domain. Depending on each child’s individual characteristics, each professional required between 15 and 60 minutes to assess the corresponding domain.”

Please refer to lines 364-371:”Due to the design of the study, the assessors were not blinded to the children’s diagnoses. However, the researcher who performed the statistical analysis was blinded. Lastly, as in previous studies [91], assessments were conducted by a multidisciplinary team experienced in evaluating children with disabilities and trained in administering the BDI-2. It should be noted that, due to the operational structure of the center, each domain of the BDI-2 was evaluated by a professional from the area most closely related to that domain (e.g., a physiotherapist evaluated the motor domain). As such, the same professional evaluated all children within the same domain”.

  1. The statistical analysis is appropriately designed and executed. ANOVA, post-hoc comparisons, and effect sizes are correctly reported. For greater clarity and accessibility, the inclusion of visual data representations—such as bar graphs or radar charts comparing DQs across domains and groups—is strongly recommended.

Answer: We thank the reviewer for this comment. Accordingly, we have included a new figure containing bar graph charts that represent post-hoc pairwise comparisons of the BDI-2 global DQ and domain-specific DQs. Please, see Figure 2 in the revised version of the manuscript.

  1. The results are well organized and statistically robust. However, the small sample size in the DLD group (n = 7) significantly limits generalizability and increases the risk of Type II error. This should be explicitly acknowledged and discussed in greater depth. Additionally, Table 1 could benefit from the inclusion of effect size measures for categorical variables (e.g., Cramér’s V).

Answer: We thank the reviewer for his/her kind suggestion. Following his/her advice, we have discussed this information in the limitations section. Additionally, we have included a new column with the effect size measure in Table 1. The partial η2 was used for age and the contingency coefficient (C) for sex since this is a 2x3 cross tabulation table. Please, see Table 1 in the revised version of the manuscript.

Please, refer to lines 345-353: “Furthermore, the small sample size in the DLD group (n = 7) substantially limits the generalizability of the results and increases the risk of a Type II error. While this is a notable limitation, current scientific literature on the developmental profiles of preschool-aged children with DLD remains sparse. In this context, the study contributes to an important area, as ensuring that children are ready to learn when they start school has become a key priority. However, the general education curriculum often does not accommodate preschool children with developmental delays [90]. Therefore, it is essential to identify developmental profiles at this stage, especially in frequently occurring medical conditions.”

  1. The discussion appropriately interprets the findings in the context of existing literature. Nevertheless, the clinical significance of statistically significant differences should be addressed more directly. The observation that children with GDD showed the highest DQs deserves deeper exploration, particularly considering the heterogeneity in outcomes reported by prior studies.

Answer: We thank the reviewer for his/her valuable suggestion. In this regard, we have acknowledged the clinical usefulness of our results. Additionally, the discussion has been expanded with regard to GDD.

Please refer to lines 241-246: “Our results showed that, compared to individuals with DS and DLD, children with GDD obtained the highest global and domain-specific DQs, scoring, on average, above 70 in some domains (personal/social and adaptive), above 80 in others (motor and communication), and 100 in the cognitive domain. Moreover, the GDD group achieved an average global DQ score of 84.25 ± 16.09, which was significantly higher than the average scores obtained by the DS (54.74 ± 8.65) and DLD (65.49 ± 16.33) groups.”

Please refer to lines 252-259:Moreover, toddlers with GDD often experience delays across multiple domains [9], making the clinical presentation of GDD heterogeneous. Notably, these delays tend to be more severe and pronounced in specific developmental areas when children with GDD present additional comorbid conditions, such as autism spectrum disorder [41,42] or intellectual disability [43,44]. For this reason, some researchers highlight the need for genetic testing to determine the underlying nature of GDD [45,46]. It is widely accepted that GDD is characterized by diverse phenotypes and a complex etiology [46].”

Please refer to lines 278-281:”Therefore, our findings highlight the need to develop early intervention strategies for communication in children with DS, as well as the importance of regular assessments to monitor progress and detect potential limitations [55], especially during this critical period.”

Please refer to lines 298-301:”In this regard, our results underscore the importance of addressing the cognitive aspects of DS through therapeutic interventions. Exercise interventions, in particular, have been shown to produce substantial enhancements in cognitive functioning in individuals with DS [66].”  

Please refer to lines 313-317: “Based on this, professionals may advise parents of children with DLD to adopt specific parenting practices—such as fostering and modelling positive relationships and supporting opportunities to practice social and adaptive skills—to help their children improve their abilities. This, in turn, could contribute to language development [78,79].”

  1. The use of language should be more precise—terms such as “more advanced developmental profile” should be replaced by specific references to the domains and comparative metrics involved. The use of the term DLD should be consistent throughout the discussion and aligned with international consensus.

Answer: We thank the reviewer for his/her valuable comments. In this regard, we have revised the document to ensure terminology. In addition, metrics have been added to specify the developmental profile in our GDD group. Additionally, a reference justifying the use of the term 'DLD' has been added to the introduction.

Please refer to lines 60-64: “DLD, by contrast, is characterized by impaired expressive and/or receptive language skills, despite normal hearing and appropriate non-verbal intelligence [5–7,12,13]. It occurs in the absence of neurological damage, structural oral anomalies, or other med-ical conditions [12–15], and is most frequently diagnosed in children under 5 years of age [16].”

Please refer to lines 241-246: “Our results showed that, compared to individuals with DS and DLD, children with GDD obtained the highest global and domain-specific DQs, scoring, on average, above 70 in some domains (personal/social and adaptive), above 80 in others (motor and communication), and 100 in the cognitive domain. Moreover, the GDD group achieved an average global DQ score of 84.25 ± 16.09, which was significantly higher than the average scores obtained by the DS (54.74 ± 8.65) and DLD (65.49 ± 16.33) groups.”

  1. The authors adequately identify key limitations, including the retrospective design and single-site data source. However, additional considerations should be incorporated:

    The absence of longitudinal data tracking change over time.

    The lack of control over potentially confounding variables (e.g., duration/intensity of intervention, socioeconomic status, level of family involvement).

These factors could have influenced the developmental outcomes reported and should be acknowledged accordingly.

Answer: We thank the reviewer for his/her valuable suggestion. We have acknowledged these limitations in line with his/her advice.

Please refer to lines 353-363:Additionally, several factors may have influenced our results. This was a retrospective analysis based solely on BDI-2 scores obtained either at the conclusion of early therapeutic intervention or just before EIC discharge. Consequently, future research should evaluate and compare children’s progress both upon entering and just before leaving the EIC to gain a more comprehensive understanding of the developmental trajectories of GDD, DLD, and DS. Moreover, our study did not explore the underlying causes of GDD in the participants. Future studies are encouraged to investigate developmental profiles in this population while considering the etiology of the disorder and other potential confounding factors that may influence children’s outcomes (e.g., sex, age, duration and intensity of the intervention, socioeconomic status, level of family involvement, family environment, and concurrent therapies).”

  1. The conclusions are consistent with the study findings. However, a more nuanced interpretation is warranted, particularly in relation to clinical application and future research directions. The authors should consider recommending multicenter, longitudinal studies tracking progress from admission to discharge in early intervention settings to further validate and expand upon these findings.

Answer: We thank the reviewer for his/her valuable suggestion. In the conclusions, we have highlighted the clinical relevance of analyzing developmental profiles in different groups with developmental disabilities and comparing them with each other. We have also provided recommendations for future studies, particularly regarding the development of multicenter and longitudinal studies.

Please refer to lines 380-392:From a clinical perspective, the findings underscore the need to implement early intervention strategies targeting communication in children with DS, along with the importance of regular assessments to monitor progress and identify potential limitations, particularly during the transition from preschool to school. Specialists are also encouraged to counsel parents of children with DLD to adopt specific behaviors aimed at enhancing their children’s social and adaptive skills, with the potential to improve communication abilities.

These findings highlight the importance of assessing developmental profiles in children with different types of developmental disabilities, as this can guide the creation of personalized intervention strategies and support parental guidance, ultimately improving outcomes for these children. Future multicenter and longitudinal studies could corroborate and expand upon these findings by tracking progress from admission to discharge in a larger sample of EIC users.”

  1. Comments on the Quality of English Language: Minor typographical errors (e.g., “toe” instead of “to each” in line 107) should be corrected.

Answer: The authors appreciate the comments of the reviewer. The typographical error has been corrected. In addition, the manuscript was reviewed by a native English speaker. Reviewers can view the certificate in the attached documents. Changes to improve the quality of the English language can be seen throughout the document.

Please refer to line 123: “to each participant”.

Reviewer 2 Report

Comments and Suggestions for Authors

The manuscript addresses a relevant and timely topic in the field of child development, offering a modest yet valuable contribution by comparing developmental profiles among preschool children diagnosed with Down Syndrome (DS), Global Developmental Delay (GDD), and Specific Language Impairment (SLI). While the retrospective study design and relatively small sample size limit the generalizability of the findings, the research is well-grounded in the current literature and holds practical implications for early intervention practices.

Abstract section is generally well structured, but its clarity could be improved by explicitly stating the research hypothesis, clearly identifying the sampling method (which is based on convenience sampling), and emphasizing the practical relevance of the differences observed between groups.

The introduction provides a solid theoretical foundation. However, it would benefit from the inclusion of more recent references (published between 2021 and 2024) to better reflect the current state of knowledge. Moreover, the scientific gap that this study aims to fill should be more clearly articulated, and the clinical and developmental distinctions between SLI and GDD should be further clarified to enhance conceptual differentiation.

In the Methods section, the procedures for administering the BDI-2 should be described in greater detail. Information such as who administered the assessments, the timing of the evaluations, and whether inter-rater reliability was ensured would strengthen the transparency and reproducibility of the study. Additionally, the authors are encouraged to discuss how potential confounding variables (e.g., sex, age, duration of intervention) were considered or controlled. The retrospective nature of the study should also be acknowledged with appropriate discussion of the inherent methodological limitations.

The results are clearly organized and supported by relevant statistical analysis. However, the interpretation of findings within the text should be expanded for greater clarity, particularly regarding the post-hoc comparisons. The flow diagram (Figure 1) is helpful, though it would be more informative if it included further details regarding participant selection and exclusion.

Discussion section is well-structured and appropriately contextualized within existing research. Nonetheless, the clinical implications of the findings could be elaborated further, especially in terms of how they might inform early intervention strategies. Additionally, the authors may wish to consider dedicating a subsection to practical recommendations. The representativeness of the sample should be more explicitly addressed, given that the data were collected from a single center.

In the Limitations section, the authors should consider explicitly mentioning the lack of longitudinal follow-up and the potential influence of unmeasured external factors, such as family environment or concurrent therapies, which may have affected the developmental outcomes.

The conclusions are concise and well aligned with the study results. To enhance their impact, the authors might reflect more directly on the clinical utility of comparative developmental profiling, particularly for tailoring individualized intervention strategies.

A minor revision is recommended, particularly to enhance methodological clarity and improve the overall quality of English expression.

Author Response

We sincerely appreciate all the comments and suggestions for improvement of the manuscript provided by Reviewers 1, 2 and 3. Below, please find our point-by-point responses to each of the questions and suggestions. The changes in the manuscript are highlighted in yellow.

Reviewer 2:

The manuscript addresses a relevant and timely topic in the field of child development, offering a modest yet valuable contribution by comparing developmental profiles among preschool children diagnosed with Down Syndrome (DS), Global Developmental Delay (GDD), and Specific Language Impairment (SLI). While the retrospective study design and relatively small sample size limit the generalizability of the findings, the research is well-grounded in the current literature and holds practical implications for early intervention practices.

  1. Abstract section is generally well structured, but its clarity could be improved by explicitly stating the research hypothesis, clearly identifying the sampling method (which is based on convenience sampling), and emphasizing the practical relevance of the differences observed between groups.

Answer: We thank the reviewer for his/her kind suggestion. Following his/her advice, we have rewritten the abstract to include the hypothesis and clarified the method of sample selection and the results.

Please refer to lines 18-19: “It was hypothesized that developmental profiles would differ depending on the type of developmental disability.”

Please refer to lines 19-24: “A cross-sectional study was conducted. Scores on the Battelle® Developmental Inventory, 2nd Edition (BDI-2) were retrieved for a non-probabilistic convenience sample of 46 children diagnosed with Down syndrome (DS) (n = 22), global developmental delay (GDD) (n = 17), and developmental language disorder (DLD) (n = 7) upon completion of an early intervention program.”

Please refer to lines 31-36: “These findings underscore the importance of early intervention strategies to improve communication in children with DS and highlight the need for regular assessments to monitor progress and identify potential limitations, particularly during the pre-school-to-school transition. Additionally, specialists should advise parents of children with DLD to adopt specific behaviors that support the development of their children’s social, adaptive, and language skills.”

  1. The introduction provides a solid theoretical foundation. However, it would benefit from the inclusion of more recent references (published between 2021 and 2024) to better reflect the current state of knowledge. Moreover, the scientific gap that this study aims to fill should be more clearly articulated, and the clinical and developmental distinctions between SLI and GDD should be further clarified to enhance conceptual differentiation.

Answer: We would like to thank the reviewer for their valuable suggestions. In this regard, we have added more recent references to the introduction section. Furthermore, we have rewritten some parts of the introduction to make the difference between GDD and DLD clearer. In addition, to provide a clearer explanation of the gap that this research aims to address, some of the introduction has been rewritten and more information has been included.

Please refer to lines 54-64: “These delays may be caused by various childhood conditions, most frequently including Down syndrome (DS), neuropsychomotor or global developmental delay (GDD), and developmental language disorder (DLD) [5]. DS arises from congenital abnormalities and is the most common aneuploid condition present at birth [5–7]. GDD is a relatively common developmental disorder for which no underlying cause is readily apparent [8,9], and it typically involves delays in at least two developmental domains [10,11]. DLD, by contrast, is characterized by impaired expressive and/or receptive language skills, despite normal hearing and appropriate non-verbal intelligence [5–7,12,13]. It occurs in the absence of neurological damage, structural oral anomalies, or other medical conditions [12–15], and is most frequently diagnosed in children under 5 years of age [16]

Please refer to lines 75-92: “This underscores the need for a comprehensive analysis and comparison of the developmental profiles of children with such disabilities. The scientific literature includes studies examining the developmental profiles of children with various medical conditions, such as severe acute malnutrition [20], middle cerebral artery infarction, periventricular infarction [21], and autism spectrum disorder [22]. Nevertheless, analyzing developmental profiles across different medical conditions is important, as children with varying conditions may respond differently to rehabilitation interventions [21]. Such an approach may offer a broader perspective on common challenges and highlight inadequacies in current intervention strategies.

However, research examining and comparing the developmental profiles of children with different developmental disabilities remains limited. Many studies focus on only one or two groups of children with disabilities, or divide children with the same diagnosis into subgroups [14–17], and only a few compare multiple groups with di-verse developmental conditions [23]. Moreover, existing studies tend to focus on a limited number of developmental areas [24–26], overlooking the importance of thoroughly evaluating multiple domains. Furthermore, most research in this field centers on newborns, school-aged children, or preadolescents [25–28], with relatively few studies involving preschool-aged children who have received therapeutic intervention in an early intervention center (EIC) [23].”

New references:

  1. Chen, L.; Wang, L.; Wang, Y.; Hu, H.; Zhan, Y.; Zeng, Z.; Liu, L. Global, Regional, and National Burden and Trends of Down Syndrome From 1990 to 2019. Front. Genet. 2022, 13, 908482, doi:10.3389/fgene.2022.908482.
  2. Aldosari, A.N.; Aldosari, T.S. Comprehensive Evaluation of the Child with Global Developmental Delays or Intellectual Disability. Clin Exp Pediatr 2024, 67, 435–446, doi:10.3345/cep.2023.01697.
  3. Sharma, A.R.; Siddiqui, M.S.; Magar, S.; Kale, A.; Nelanuthala, M.; Singh, S.P. The Etiological Profile of Global Developmental Delay at a Tertiary Care Hospital in India: An Observational Study. Cureus 2023, doi:10.7759/cureus.41066.
  4. Lancaster, H.S.; Camarata, S. Reconceptualizing Developmental Language Disorder as a Spectrum Disorder: Issues and Evidence. Intl J Lang & Comm Disor 2019, 54, 79–94, doi:10.1111/1460-6984.12433.
  5. Bishop, D.V.M.; Snowling, M.J.; Thompson, P.A.; Greenhalgh, T.; and the CATALISE2 consortium Phase 2 of CATALISE: A Multinational and Multidisciplinary Delphi Consensus Study of Problems with Language Development: Terminology. Child Psychology Psychiatry 2017, 58, 10681080, doi:10.1111/jcpp.12721.
  6. Calder, S.D.; BrennanJones, C.G.; Robinson, M.; Whitehouse, A.; Hill, E. The Prevalence of and Potential Risk Factors for Developmental Language Disorder at 10 Years in the Raine Study. J Paediatrics Child Health 2022, 58, 20442050, doi:10.1111/jpc.16149.
  7. Koul, A.M.; Ahmad, F.; Bhat, A.; Aein, Q.; Ahmad, A.; Reshi, A.A.; Kaul, R.-R. Unraveling Down Syndrome: From Genetic Anomaly to Artificial Intelligence-Enhanced Diagnosis. Biomedicines 2023, 11, 3284, doi:10.3390/biomedicines11123284.
  8. Saleem, J.; Zakar, R.; Mushtaq, F.; Bukhari, G.M.J.; Fischer, F. Comparative Analysis of Developmental Profile between Normal and Severe Acute Malnourished Under-Five Children in Pakistan: A Multicentre Cross-Sectional Study. BMJ Open 2021, 11, e048644, doi:10.1136/bmjopen-2021-048644.
  9. Fehlings, D.; Krishnan, P.; Ragguett, R.M.; Campbell, C.; Gorter, J.W.; Hunt, C.; Kawamura, A.; Kim, M.; McCormick, A.; Mesterman, R.; et al. 95 A Comparison of the Developmental Profiles of Individuals with Hemiplegic Cerebral Palsy Associated with Middle Cerebral Artery and Periventricular Venous Infarctions. Paediatrics & Child Health 2019, 24, e36–e37, doi:10.1093/pch/pxz066.094.
  10. Roizen, N.; Friedman, S.; Vanderbilt, D.; Cacia, J.; Fussell, J.; Hansen, R.; Hofer, J.; Sideridis, G.; Stein, R.E.K.; Barbaresi, W. Developmental Profiles of Young Children Referred for Concern for Autism Spectrum Disorder: DBPNet Study. J Autism Dev Disord 2025, doi:10.1007/s10803-025-06777-0.
  11. In the Methods section, the procedures for administering the BDI-2 should be described in greater detail. Information such as who administered the assessments, the timing of the evaluations, and whether inter-rater reliability was ensured would strengthen the transparency and reproducibility of the study. Additionally, the authors are encouraged to discuss how potential confounding variables (e.g., sex, age, duration of intervention) were considered or controlled. The retrospective nature of the study should also be acknowledged with appropriate discussion of the inherent methodological limitations.

Answer: The authors are grateful for the reviewer's comments. Following the reviewer's recommendation and to clarify the Methods section, we have consulted with the early intervention center and expanded the information in the manuscript. We have also acknowledged the limitations of the retrospective analysis in the limitations section.

Please refer to lines 138-145: “In the collaborating EIC, prior to admission to primary school, the BDI-2 was administered by members of the early intervention team, who were experienced in assessing children with disabilities and trained in BDI-2 administration. Specifically, a neuro-psychologist assessed the personal/social and cognitive domains, a physiotherapist the motor domain, a speech and language therapist the communication domain, and an occupational therapist the adaptive domain. Depending on each child’s individual characteristics, each professional required between 15 and 60 minutes to assess the corresponding domain.”

Please refer to lines 359-363: “Future studies are encouraged to investigate developmental profiles in this population while considering the etiology of the disorder and other potential confounding factors that may influence children’s outcomes (e.g., sex, age, duration and intensity of the intervention, socioeconomic status, level of family involvement, family environment, and concurrent therapies).”

Please refer to lines 353-358: “Additionally, several factors may have influenced our results. This was a retrospective analysis based solely on BDI-2 scores obtained either at the conclusion of early therapeutic intervention or just before EIC discharge. Consequently, future research should evaluate and compare children’s progress both upon entering and just before leaving the EIC to gain a more comprehensive understanding of the developmental trajectories of GDD, DLD, and DS.”

  1. The results are clearly organized and supported by relevant statistical analysis. However, the interpretation of findings within the text should be expanded for greater clarity, particularly regarding the post-hoc comparisons. The flow diagram (Figure 1) is helpful, though it would be more informative if it included further details regarding participant selection and exclusion.

Answer: We would like to thank the reviewer for their valuable suggestions. In this regard, we have added more information in the flow diagram regarding participant selection and exclusion. Please, see figure 1. In addition, we have added another figure to clarify post-hoc comparisons. Please, see figure 2. Finally, the findings have been expanded for greater clarity at the beginning of the discussion.

Please refer to lines 241-246: “Our results showed that, compared to individuals with DS and DLD, children with GDD obtained the highest global and domain-specific DQs, scoring, on average, above 70 in some domains (personal/social and adaptive), above 80 in others (motor and communication), and 100 in the cognitive domain. Moreover, the GDD group achieved an average global DQ score of 84.25 ± 16.09, which was significantly higher than the average scores obtained by the DS (54.74 ± 8.65) and DLD (65.49 ± 16.33) groups.”

  1. Discussion section is well-structured and appropriately contextualized within existing research. Nonetheless, the clinical implications of the findings could be elaborated further, especially in terms of how they might inform early intervention strategies. Additionally, the authors may wish to consider dedicating a subsection to practical recommendations. The representativeness of the sample should be more explicitly addressed, given that the data were collected from a single center.

Answer: Thank you for your valuable comment. Recommendations have been added throughout the discussion, and the clinical relevance of the results has been highlighted. The limitations of the sample size have also been acknowledged, and the development of future multicenter studies encouraged.

Please refer to lines 278-281: “Therefore, our findings highlight the need to develop early intervention strategies for communication in children with DS, as well as the importance of regular assessments to monitor progress and detect potential limitations [55], especially during this critical period.”

Please refer to lines 298-301: “In this regard, our results underscore the importance of addressing the cognitive aspects of DS through therapeutic interventions. Exercise interventions, in particular, have been shown to produce substantial enhancements in cognitive functioning in individuals with DS [66].”  

Please refer to lines 313-317: “Based on this, professionals may advise parents of children with DLD to adopt specific parenting practices—such as fostering and modelling positive relationships and sup-porting opportunities to practice social and adaptive skills—to help their children improve their abilities. This, in turn, could contribute to language development [78,79].”

Please refer to lines 342-352: “The participants were drawn from a single center, which may have introduced selection bias. However, the requirements for access to an EIC remain consistent across all Spanish centers. Furthermore, the small sample size in the DLD group (n = 7) substantially limits the generalizability of the results and increases the risk of a Type II error. While this is a notable limitation, current scientific literature on the developmental profiles of preschool-aged children with DLD remains sparse. In this context, the study contributes to an important area, as ensuring that children are ready to learn when they start school has become a key priority. However, the general education curriculum often does not accommodate preschool children with developmental delays [90]. Therefore, it is essential to identify developmental profiles at this stage, especially in frequently occurring medical conditions.”

  1. In the Limitations section, the authors should consider explicitly mentioning the lack of longitudinal follow-up and the potential influence of unmeasured external factors, such as family environment or concurrent therapies, which may have affected the developmental outcomes.

Answer: We thank the reviewer for his/her insightful comment. In this regard, we have acknowledged the limitations of the study, namely the absence of longitudinal follow-up of the participants and control for possible confounding variables.

Please refer to lines 353-363: “Additionally, several factors may have influenced our results. This was a retrospective analysis based solely on BDI-2 scores obtained either at the conclusion of early therapeutic intervention or just before EIC discharge. Consequently, future research should evaluate and compare children’s progress both upon entering and just before leaving the EIC to gain a more comprehensive understanding of the developmental trajectories of GDD, DLD, and DS. Moreover, our study did not explore the underlying causes of GDD in the participants. Future studies are encouraged to investigate developmental profiles in this population while considering the etiology of the disorder and other potential confounding factors that may influence children’s outcomes (e.g., sex, age, duration and intensity of the intervention, socioeconomic status, level of family involvement, family environment, and concurrent therapies).”

  1. The conclusions are concise and well aligned with the study results. To enhance their impact, the authors might reflect more directly on the clinical utility of comparative developmental profiling, particularly for tailoring individualized intervention strategies.

Answer: We thank the reviewer for his/her valuable suggestion. In the conclusions, we have highlighted the clinical relevance of analyzing developmental profiles in different groups with developmental disabilities and comparing them with each other. In this regard, we have emphasized in the conclusions section the importance of comparing developmental profiles to enable more tailored interventions.

Please refer to lines 387-392: “These findings highlight the importance of assessing developmental profiles in children with different types of developmental disabilities, as this can guide the creation of personalized intervention strategies and support parental guidance, ultimately improving outcomes for these children. Future multicenter and longitudinal studies could corroborate and expand upon these findings by tracking progress from admission to discharge in a larger sample of EIC users.”

  1. A minor revision is recommended, particularly to enhance methodological clarity and improve the overall quality of English expression.

Answer: The authors appreciate the comments of the reviewer. The manuscript was reviewed by a native English speaker. Reviewers can view the certificate in the attached documents. Changes to improve the quality of the English language can be seen throughout the document.

Reviewer 3 Report

Comments and Suggestions for Authors

This study presents a useful comparison of developmental profiles in preschool children with Down Syndrome, Global Developmental Delay, and Specific Language Impairment using the BDI-2. The findings reveal significant differences across groups, with GDD children performing relatively better.

The topic is very interesting, and the fact that a comprehensive assessment tool was used strengthens the scientific approach. However, it would be important to justify the small sample size. Additionally, the authors should elaborate more on the clinical implications of these findings.

Overall, I believe the study should be published after minor revisions.

Author Response

We sincerely appreciate all the comments and suggestions for improvement of the manuscript provided by Reviewers 1, 2 and 3. Below, please find our point-by-point responses to each of the questions and suggestions. The changes in the manuscript are highlighted in yellow.

Reviewer 3

This study presents a useful comparison of developmental profiles in preschool children with Down Syndrome, Global Developmental Delay, and Specific Language Impairment using the BDI-2. The findings reveal significant differences across groups, with GDD children performing relatively better.

  1. The topic is very interesting, and the fact that a comprehensive assessment tool was used strengthens the scientific approach. However, it would be important to justify the small sample size. Additionally, the authors should elaborate more on the clinical implications of these findings.

Answer: We thank the reviewer for his/her valuable comment. Following it, the clinical relevance of the results has been highlighted. The limitations of the sample size have also been acknowledged.

Please refer to lines 278-281: “Therefore, our findings highlight the need to develop early intervention strategies for communication in children with DS, as well as the importance of regular assessments to monitor progress and detect potential limitations [55], especially during this critical period.”

Please refer to lines 298-301: “In this regard, our results underscore the importance of addressing the cognitive aspects of DS through therapeutic interventions. Exercise interventions, in particular, have been shown to produce substantial enhancements in cognitive functioning in individuals with DS [66].”  

Please refer to lines 313-317: “Based on this, professionals may advise parents of children with DLD to adopt specific parenting practices—such as fostering and modelling positive relationships and sup-porting opportunities to practice social and adaptive skills—to help their children improve their abilities. This, in turn, could contribute to language development [78,79].”

Please refer to lines 342-352: “The participants were drawn from a single center, which may have introduced selection bias. However, the requirements for access to an EIC remain consistent across all Spanish centers. Furthermore, the small sample size in the DLD group (n = 7) substantially limits the generalizability of the results and increases the risk of a Type II error. While this is a notable limitation, current scientific literature on the developmental profiles of preschool-aged children with DLD remains sparse. In this context, the study contributes to an important area, as ensuring that children are ready to learn when they start school has become a key priority. However, the general education curriculum often does not accommodate preschool children with developmental delays [90]. Therefore, it is essential to identify developmental profiles at this stage, especially in frequently occurring medical conditions.”

Overall, I believe the study should be published after minor revisions.